# PL Tunable GaN Nanoparticles Synthesis through Femtosecond Pulsed Laser Ablation in Different Environments

**DOI:** 10.3390/nano10030439

**Published:** 2020-02-29

**Authors:** Juan Hao, Sijia Xu, Bingrong Gao, Lingyun Pan

**Affiliations:** 1College of Electronic Science and Engineering, Jilin University, Changchun 130012, China; haojuan16@mails.jlu.edu.cn (J.H.); sjxu18@mails.jlu.edu.cn (S.X.); 2College of physics, Jilin University, Changchun 130012, China

**Keywords:** GaN-NPs, fs-PLAL, tunable PL properties, ablation environments

## Abstract

The tunable photoluminescence (PL) property is very important for gallium nitride (GaN) nanoparticles in the application of ultraviolet and blue optoelectronic devices, while conventional methods are not so satisfactory that alternative methods for preparing GaN nanoparticles should be studied. In this paper, ultra-small and well dispersed GaN nanoparticles are fabricated through femtosecond pulse laser ablation in air, water and ethanol. For the PL spectra of GaN nanoparticles, there are no shifts in air, red shifts in water and blue shifts in ethanol compared with the intrinsic PL spectra of bulk GaN. The X-ray photoelectron spectroscopy (XPS) results demonstrate that the various PL spectra can be due to the different components inside the GaN nanoparticles, which not only have effect on the PL emissions, but also greatly influence the intensity of PL. This study validates that the ablation environment has a great adjustable effect on the properties of GaN nanoparticles.

## 1. Introduction

Material science is a discipline designed to meet specific application requirements [1]. It regulates the properties of materials by altering their composition or morphology, and nanomaterials are no exception [2]. The nanomaterials are featured with artificial design and adjustable synthesis. When the size of semiconductor materials is less than the exciton Bohr radius, the band structure will change due to the quantum confinement effect. As a result, unique optical properties that cannot be found in bulk gallium nitride (GaN) will emerge and these properties endow many more potential applications in optoelectronic devices and other many fields to nanomaterials [3]. Ⅲ‒Ⅴ group nanomaterials have wide and direct band-gap, which has aroused increasing interest in recent years [4,5]. GaN has a band-gap of 3.4 eV at 300 K, and the band-gap edge photoluminescence (PL) corresponds to the ultraviolet (UV) region. GaN-based materials not only can be used in blue light transmitters, but also have important applicable values in high-intensity, high-power, high-temperature and high-frequency optoelectronic devices [6,7]. Previous studies have shown that GaN nanoparticles (GaN-NPs) not only inherited the excellent properties of crystalline materials, but also opened up a new application prospect, such as good gas sensitivity, especially for H_2_ and H_2_S [8]. Also, they have unique advantages as luminescence probes in biological imaging, because their absorption spectra can be adjustable from UV to near-infrared and narrow luminescence spectra with accessible range wavelength that can be compared with traditional organic dyes or fluorescent proteins [9]. More importantly, the PL of GaN-NPs will not bleach, which is very suitable for in vivo studies [10]. In addition, GaN-NPs have been candidate as the single photon source for optical switch devices [11]. All of these accelerate the research into GaN-NPs. 

The earliest method for synthesizing GaN-NPs was laser-assisted MBE (Molecular Beam Epitaxy), which has high adjustability but is hard to separate NPs from the substrate [12]. Whereas chemical methods can prepare GaN-NPs with high crystallinity and uniform size distribution, Chen et al. synthesized GaN-NPs with 50 nm size by the reaction of gallium chloride (GaCl_3_) and lithium nitride (Li_3_N) at 290 °C, and TOPO (Trioctylphosphine Oxide) was added to prevent particle agglomeration [13]. But the size of 50 nm is far beyond the exciton Bohr radius which is 11 nm for GaN [14]. Although ultra-small GaN-NPs (3~4 nm) have been obtained by chemical means, their properties cannot be regulated and the by-products are complex [15]. Through the capture effect of Ga metal gas and N_2_ non-thermal plasma, You et al successfully synthesized pure GaN-NPs with a size distribution of 10–40 nm but the yield is rather barren [16], as the thermal plasma method is suffering the same problem [17]. In order to overcome the current difficulties, pulsed laser ablation methods which take into account properties control, high yields and finely purity products are generated [18]. It was found that only 3,000 laser pulses in a gas environment can successfully prepare 1 cm^2^ of GaN-NPs, which created confidence in large-scale yield [19]. However, GaN-NPs prepared by laser ablation in a gas environment are easily agglomerated into large particles, which is not conducive to their application. Subsequently, the preparation of GaN-NPs by pulsed laser ablation in liquid (PLAL) was reported. Through femtosecond laser ablation in ethanol, the GaN-NPs with 4.2 ± 1.9 nm size distribution and 310 nm PL emission were synthesized successfully, which satisfied small size, uniform distribution and blue-shifted PL simultaneously [20]. So far, PLAL has been recognized as a green and one-step method to obtain ultra-small sized and highly pure GaN-NPs. Nevertheless, the correlated reports are rather scarce.

In this paper, the GaN-NPs are prepared by femtosecond pulsed laser ablation GaN film in air, deionized water and ethanol. The main purpose is to explore the property changes of GaN-NPs in different environments, including the particles’ size, particles’ components and thus the optical properties. This results show that fs-PLAL can prepare PL tunable GaN-NPs by varying the ablation environment, and provide a possibility for controllable GaN-NPs to be widely adjusted and applied.

## 2. Materials and Methods 

### 2.1. Materials

The target materials are c-GaN films with sapphire substrate which are grown by CVD. The thickness of GaN film is 2 μm. The other materials, including water and ethanol, used in experiments are analytical grade without any treatments.

### 2.2. Experiments

Figure 1a is the scheme of fs-PLAL. GaN-NPs were fabricated by ablating GaN film in air, water and ethanol using 343 nm fs-laser (Light-Conversion Pharos, Vilnius, Lithuania) with 130 fs pulse width and 90 kHz repetition rate. The target material was cleaned by ultrasonics with ethanol and deionized water and then adhered on the interior back of quartz cuvette. The laser passed through the aperture (A1), beam expander (L1 and L2), all-reflection mirror (M1), field len (L3, *f* =10 cm) and was focused by L3 onto the GaN film. The ablation lasted one hour for every sample. The ablation powers used were different in various environments because the absorption of 343 nm light in air, water and ethanol is quite different. In order to avoid the damage in sapphire during the ablation process, extensive experiments have been conducted to confirm the energy densities per pulse used in experiments, which were 14.1 J∙cm^-2^/pulse in air, 97.3 J∙cm^-2^/pulse in water and 338.2 J∙cm^-2^/pulse in ethanol. During ablation in air, in order to prevent the agglomeration in air, deionized water was added into cuvette, and the liquid level was closed to the lowest point of the laser focus scanning, as displayed in Figure 1b. When scanning in deionized water and ethanol, the cuvette was filled with the corresponding liquid to make the laser pass through the liquid medium and was focused on the surface of the GaN film, as displayed in Figure 1c. After ablation, the GaN colloid was collected and subjected to ultrasonics for 40 minutes. They were then centrifuged at a high speed of 12,000 rpm/min for 20 minutes, and the supernatant liquid was used for tested. The products were named GaN-NPs@air, GaN-NPs@water and GaN-NPs@ethanol, respectively.

### 2.3. Characterizations

The GaN-NPs’ morphology and size distribution were studied by transmission electron microscopy (TEM) and high-resolution TEM (HRTEM, JEOL JEM-2100F, Tokyo, Japan). To confirm their crystalline phase structure, the Raman spectra were characterized by JOBIN YVON T64000 (Paris, France). The X-ray photoelectron spectra (XPS) were conducted by ESCALAB250 (Thermo Fisher Scientific, Waltham, MA, USA) to analyze atomic compositions and bonds by dropping the GaN colloid onto Si substrate. Their optical properties were studied by ultraviolet-visible (UV-Vis) absorption spectra (UV3600 UV-Vis-NIR spectrophotometer, SHIMADZU, Kyoto, Japan), PL and excitation spectra (F4600 fluorescence spectrophotometer, HITACHI, Tokyo, Japan). 

## 3. Results

### 3.1. Morphology and Structural Properties of GaN-NPs

Figure 2A–F are the TEM and HRTEM images of GaN-NPs@air, GaN-NPs@water and GaN-NPs@ethanol. As can be seen from Figure 2A–C, the GaN-NPs prepared by fs-PLAL have good dispersion, especially the GaN-NPs obtained by ablation in air, which are allowed to disperse into water while sputtering, thus solving the problem of GaN-NPs’ aggregation in air. The HRTEM insets show that the crystal lattice structures of the GaN-NPs are consistent and all have good crystallinity. All the face spacings are measured to be about 0.24 nm, corresponding to the (111) crystal face spacing of c-GaN lattice structure. Moreover, the size distributions are relatively uniform, and the shapes of GaN-NPs are all round or nearly round as shown in Figure 2D–F. The size distributions show that the average sizes of GaN-NPs@air, GaN-NPs@water and GaN-NPs@ethanol are 3.2 nm (Figure 2G), 2.7 nm (Figure 2H) and 2.3 nm (Figure 2I), respectively. Furthermore, the uniformity of GaN-NPs@water is the best. Therefore, compared with other preparation technologies, fs-PLAL can produce ultra-small GaN-NPs (< 4 nm) with rather uniform distribution, which is beneficial to study the effect of size on their various properties. It can be seen from the results that a certain regulatory effect on the size of GaN-NPs have been formed by the ablation environment which is closely related to the formation and growth of NPs. In the air, before being quenched by sputtering into water, the growth rate of GaN-NPs on GaN-air interface is fast due to the less confinement of air on the laser-induced plasma, which results in larger NPs. However, in water, the confinement on the laser-induced plasma is more intense due to the relatively large viscosity of water and the cooling time is short due to the large thermal conductivity of water, which render the growth rate of particles relatively slow and the growth time short [21]. Thus the size of GaN-NPs@water (2.7 nm) is smaller than that in GaN-NPs@air (3.2 nm). Based on the above theory, for the confinement of ethanol on the laser-induced plasma is smaller than that of water, the particles size in ethanol should be larger than GaN-NPs@water. However, because the ablation efficiency of 343 nm laser in ethanol is much lower than in water or air, the laser-induced plasma density is so small that leads to smaller NPs (2.3 nm).

In order to further confirm the structure of prepared GaN-NPs, Raman spectra were performed as shown in Figure 3. The Raman peaks at 568.4 cm^−1^ and 735.6 cm^−1^ are originated from E_2_ and A_1_ modes of GaN, and thus confirm the products are GaN-NPs. Compared to the GaN film, all the GaN-NPs’ E_2_ peaks blue shift and the widths of A_1_ peaks are broadened. That is because the GaN-NPs’ sizes are less than the exciton Bohr radius of GaN, and then phonon modes will blue shift as a result of the quantum confinement effect [22]. It should be pointed out that the A_1_ mode of GaN-NPs@air is very weak, possibly because their crystal structures are not perfect.

In order to confirm the differences in the elements and bonding conditions inside different GaN-NPs, X-ray photoelectron spectroscopy (XPS) characterization was conducted. As shown by Figure 4, N1s, Ga2p and Ga3d were tested respectively. According to the XPS binding energy standard table, the elements and corresponding bonds are determined. For N element, the binding energy at 398.4eV comes from Ga–N, while the binding energy at 407.2eV comes from N–O, and both peaks are evident in Figure 4a. In Figure 4b, the XPS signal peak of Ga2p is composed of Ga2p1 and Ga2p3, which located at 1144.7 eV and 1117.8 eV. The Ga2p3 peak contains two different binding energies which locate at 1117.8 eV for Ga–N and 1120 eV for Ga–O respectively, which indicates that the GaN-NPs@air contains a certain proportion of O atoms. The O atoms participate in the formation of crystal structure, that is why the Raman signal of A_1_ mode almost disappears for GaN-NPs@air. For GaN-NPs@water and GaN-NPs@ethanol, their N1s (397.4 ev), Ga2p1 (1144.7 ev) Ga2p3 (1117.4 ev) and Ga3d (20.1 ev) all have a single binding energy peak, so it can be conclude that GaN-NPs prepared in water and ethanol are likely to be free of any impurities. The ratios of Ga and N atoms in three kinds of GaN-NPs were calculated through the peak integral area of each element by XPS spectra and the XPS sensitivity factors of each atom are listed in Table 1. For GaN-NPs@air, Ga atoms account for 63%, while N element accounts for 37%. It is obvious that the ratio of Ga:N is not 1:1. That is because O atoms can provide common electron pairs to replace a part of N element (O_N_) or the presence of N vacancies (V_N_), as well as both exist. For GaN-NPs@water, Ga:N is 46:54 and 58:42 for GaN-NPs@ethanol. The mismatch of Ga and N atoms is like to result in Ga vacancy (V_Ga_) or V_N_, which is integrated into accounts that there are no other atoms in both kinds of NP.

### 3.2. Optical Properties of GaN-NPs

As a wide band-gap semiconductor material, the optical properties of GaN-NPs are very important in application of various devices. Figure 5a–c are the PL spectra of GaN-NPs@air, GaN-NPs@water and GaN-NPs@ethanol, in which the PL bands do not shift with the excitation wavelength altering, while the PL intensity varies. The PL peaks of GaN-NPs@air are distributed in the ultraviolet region (335–360 nm) which gradually redshifts as the excitation wavelengths increase and the PL intensity becomes the strongest at 347 nm excited by 310 nm. Usually, the phenomenon that the PL peaks of NPs move with the excitation wavelength is always caused by poor uniformity of the NPs’ size, which is validated by Figure 2G. The PL spectra of GaN-NPs@water are mainly distributed between 420 nm and 440 nm, and there are two obvious PL peaks at 425 nm and 440 nm in their spectra. The PL peaks of GaN-NPs@ethanol are at about 334 nm, and the intensity reaches the maximum at 300 nm excitation. On the whole, it can be concluded that these three kinds of GaN-NPs have completely different PL properties, and their PL spectra are distributed in the ultraviolet or blue light regions as revealed in Figure 5d. Even more interesting, PL intensities of GaN-NPs@ethanol overall are higher than that of the other two as exposed in Figure 5e. As analyzed above, the PL properties of GaN-NPs, including PL peak and intensity, can be regulated by selecting an ablation environment utilizing fs-PLAL technology. These results have very important guiding significance for the synthesis of GaN NPs with adjustable PL properties.

Figure 6a shows the UV-Vis absorption spectra of three kinds of GaN-NPs. The absorption trend of GaN-NPs@water is almost same as that of GaN-NPs@air with about 400 nm cut-off wavelength, except that GaN-NPs@water has a shoulder at 360 nm. The cut-off wavelength of GaN-NPs@ethanol is at about 320 nm, and there are two obvious absorption peaks at 268 nm and 220 nm. Obviously, GaN-NPs@ethanol has a higher cut-off energy, which indicates that the large band-gap leads to a blue-shift of the PL peak (334 nm) compared with intrinsic PL peak of bulk GaN (360 nm). 

The excitation spectra in Figure 6b–d all display two excitation peaks for every kind of GaN-NPs at different PL wavelengths. At luminescence 330 nm, 340 nm and 350 nm for GaN-NPs@air, the weaker one locates at 245 nm, while the stronger ones change from 300 nm to 315 nm depending on the PL emissions revealed by Figure 6b. For GaN-NPs@water’ PL at 425 nm and 440 nm, two excitation peaks locating at 290 nm and 374 nm are displayed in Figure 6c. For GaN-NPs@ethanol, Figure 6d depicts that both excitation peaks located at 245 nm and 305 nm are independent on the PL emissions. However, the excitation intensity of 245 nm is greatly weakened compared with that of 305 nm.

## 4. Discussion

On the whole, the ablation environment is responsible for the formation and growth of GaN-NPs, and regulates their PL spectra. The PL emissions of GaN-NPs prepared in air and ethanol both distribute in the ultraviolet region. The difference is that the PL emission of GaN-NPs@air depends on the excitation wavelength (Figure 5a), while that of GaN-NPs@ethanol is independent on the excitation wavelength (Figure 5c). Seen from the UV-Vis absorption spectra, the absorption cut-off wavelength of GaN-NPs@ethanol is shorter than that of both GaN-NPs@air and GaN-NPs@water. This is according to *Eg* = *hc*/*λc*, where *h* is the Planck constant, *λc* is the absorption cut-off wavelength, *c* is the speed of light, and *Eg* is band-gap, a shorter *λc* corresponds to a wider band-gap. It is assumed that if the PL emission is completely dependent on the band-gap, the PL spectra of GaN-NPs@air should be theoretically close to that of GaN-NPs@water because of their similar *λc*. However, it is not the case that the experiment results in turn disprove the assumption. The previous studies showed that the ultraviolet PL of GaN was caused by the radiation recombination of shallow DAP (donor-accepter pairs), in which the O_N_ can be the shallow donor [23]. As concluded from XPS results, O_N_ does exist in GaN-NPs@air. Therefore, GaN-NPs@air’s ultraviolet PL is probably caused by O_N_, which are further validated by the relatively weak PL intensity of GaN-NPs@air, because DAP radiation recombination efficiency is rather low. While the PL intensity of GaN-NPs@ethanol is much higher than that of GaN-NPs@air, so the PL at 334 nm of GaN-NPs@ethanol originated from their band-edge luminescence. GaN-NPs@water has two emission peaks, which are distributed in the blue light region. Compared with the PL spectrum of bulk GaN, the redshift may be caused by the Ga vacancy (V_Ga_) or more complex lattice structure related to it. Moreover, the luminescence efficiency caused by V_Ga_ is relatively weak [24,25].

Additionally, it is found that there are different contributions in all the PL spectra and the relative intensity is different. The PL spectra in Figure 5a show a rather symmetrical shape at short wavelength excitation (260–290 nm) which can be due to wide excitation as a result of high excitation energy. As the excitation wavelength shifts to longer wavelength, the photon energy is too low to excite small-sized GaN-NPs. As a result, the contribution of short wavelength emission to PL decreases and long wavelength emission becomes obvious. Compared with GaN-NPs@air, the shape change and peaks shift are much less, which are shown in Figure 5b. This is probably due to the good uniformity of size and few non-eigenstate emissions in GaN-NPs@water. Nevertheless, there are two obvious blue PL peaks and their relative intensities are almost unchanged. Therefore, the two eigenstate emissions may originate from two kinds of emission center. This just illustrates that there is not only V_Ga_ inside GaN-NPs@water, but also another related structural defect causing blue light emission. In terms of GaN-NPs@ethanol, there are two shoulders on either side of 334 nm peak in Figure 5c, which is mainly because the different sized GaN-NPs all contribute to the PL spectra. Moreover, the intensities of the two shoulders alter with the excitation wavelength, especially the right shoulder’s intensity which gradually increases, which can be explained by the photoluminescence theory. Namely, the intensity of short wavelength emission will decrease and the long emission increases as the excitation wavelength becomes longer. Compared to GaN-NPs@air, the PL emission of GaN-NPs@ethanol always keeps the emission peak barely moving, which mainly depends on the more uniform size distribution of GaN-NPs@ethanol.

## 5. Conclusions

In conclusion, well-dispersed and ultra-small GaN-NPs with adjustable size and components are obtained by fs-PLAL in air, water and ethanol and the uniformity of GaN-NPs prepared in liquid is better. The PL emissions of GaN-NPs@ethanol and GaN-NPs@air locate at the UV region, and that of GaN-NPs@water locates in the blue region. Comprehensive discussions suggest that different PL mechanisms inside each kind of GaN-NPs and the size distribution are responsible for the change in PL emission and the difference in PL intensity. It is necessary to emphasize that the PL intensity of GaN-NPs@ethanol is the strongest compared with the other two. Therefore, it is reasonable to believe that fs-PLAL technology has great potential as a viable alternative to the conventional methods in the synthesis of GaN-NPs and regulation of their properties through selecting appropriate liquid environments.

## Figures and Tables

**Figure 1 nanomaterials-10-00439-f001:**
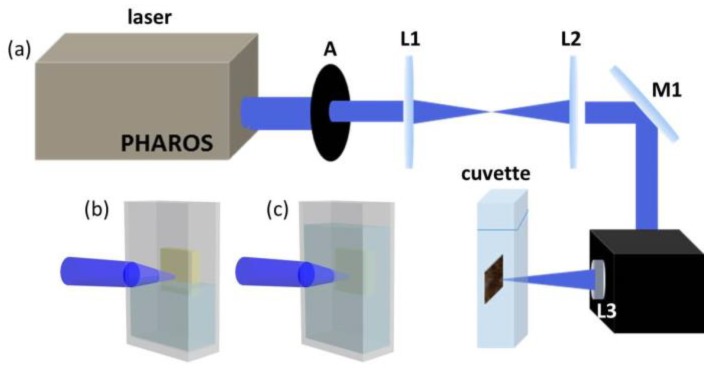
(**a**) The scheme of fs-PLAL (pulsed laser ablation in liquid); the relative position between the laser’s focus and liquid level in (**b**) air, (**c**) water and ethanol.

**Figure 2 nanomaterials-10-00439-f002:**
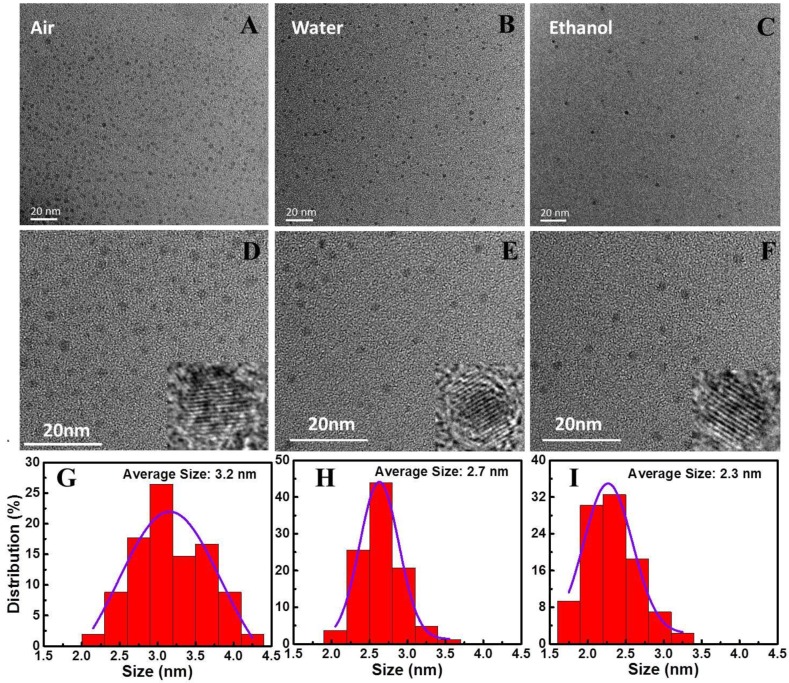
The transmission electron microscopy (TEM) images of GaN-NPs@air (**A**,**D**), GaN-NPs@water (**B**,**E**) and GaN-NPs@ethanol (**C**,**F**), the corresponding insets are their high-resolution TEM (HRTEM) images, and the corresponding size distribution are displayed in (**G**–**I**), respectively.

**Figure 3 nanomaterials-10-00439-f003:**
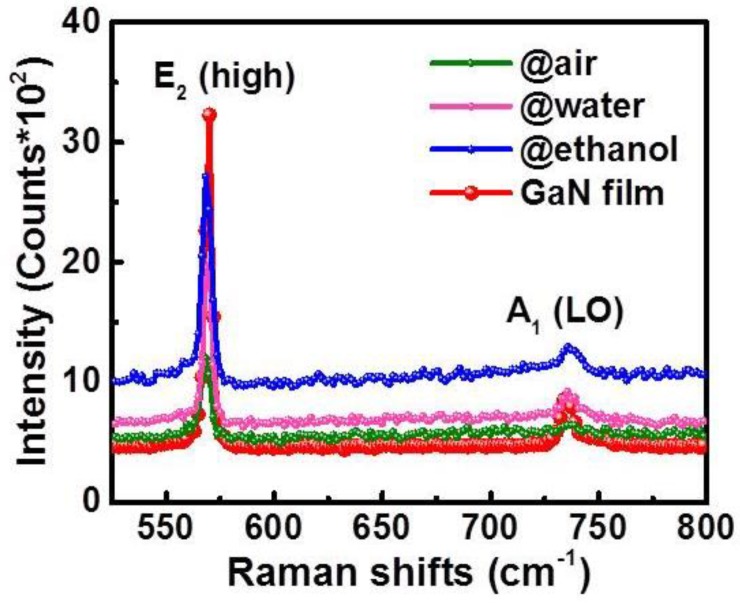
The Raman spectra of GaN-NPs@air, GaN-NPs@water, GaN-NPs@ethanol and GaN film.

**Figure 4 nanomaterials-10-00439-f004:**
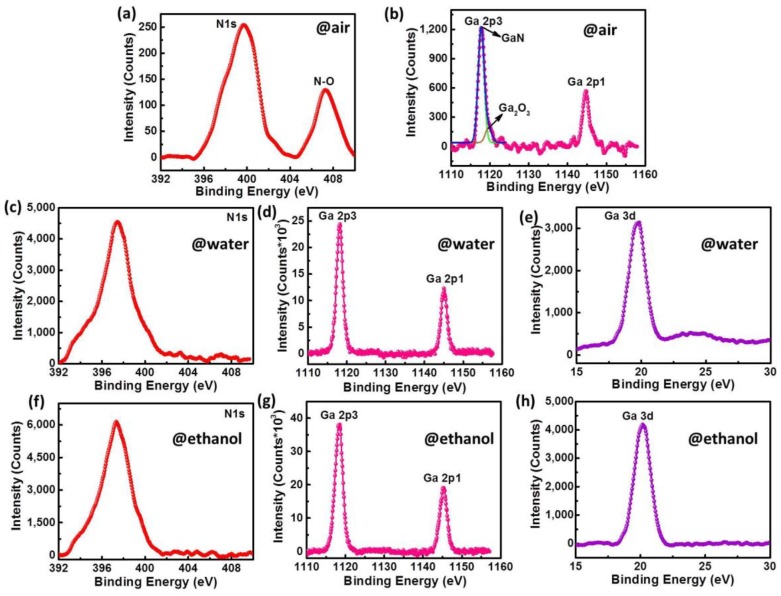
X-ray photoelectron spectroscopy (XPS) spectra of GaN-NPs: (**a**,**b**) from N1s and Ga2p of GaN-NPs@air; (**c**–**e**) are from N1s, Ga2p and Ga3d in GaN-NPs@ water; (**f**–**h**) are respectively from N1s, Ga2p and Ga3d of GaN-NPs@ethanol.

**Figure 5 nanomaterials-10-00439-f005:**
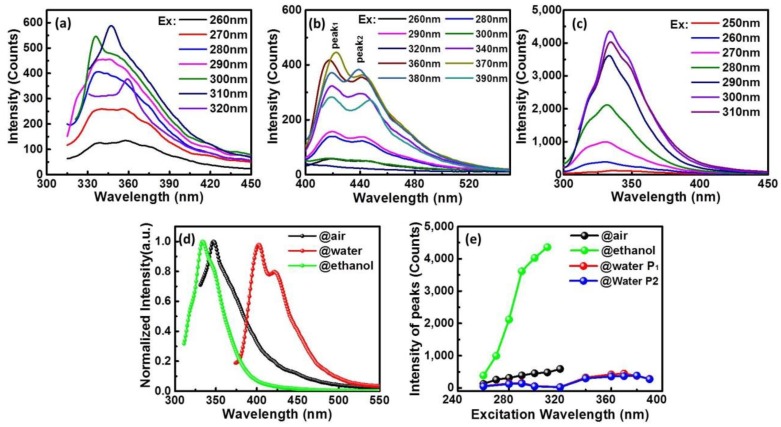
The photoluminescence (PL) spectra of (**a**) GaN-NPs@air, (**b**) GaN-NPs@water and (**c**) GaN-NPs@ethanol with different excitation wavelength; (**d**) the maximum PL emissions and (**e**) the emissions’ intensity of three kinds of GaN-NPs.

**Figure 6 nanomaterials-10-00439-f006:**
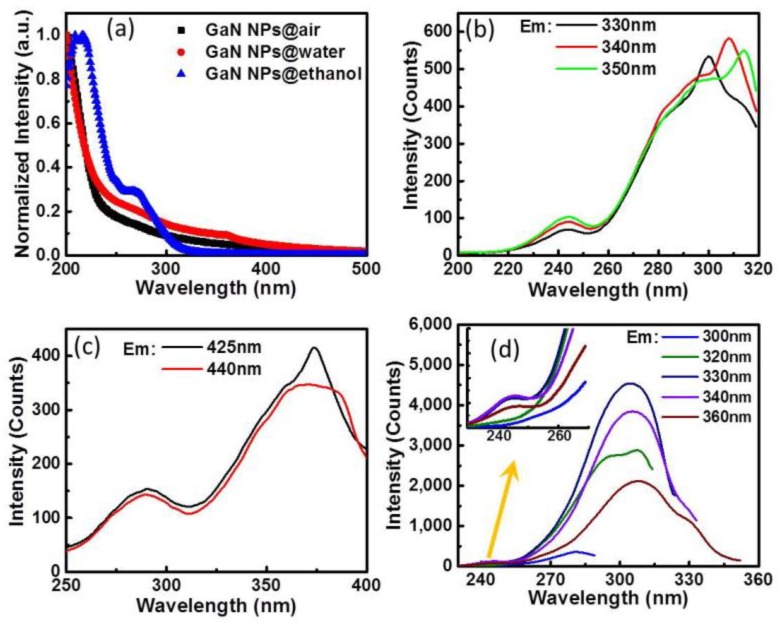
(**a**) The ultraviolet-visible (UV-Vis) absorption spectra of the three kinds of GaN-NPs; the excitation spectra of (**b**) GaN-NPs@air, (**c**) GaN-NPs@water and (**d**) GaN-NPs@ethanol.

**Table 1 nanomaterials-10-00439-t001:** The contents of Ga and N elements in GaN-NPs@air, GaN-NPs@water and GaN-NPs@ethanol.

GaN NPs	Ga2p				N1s			
	Sensitivity factor	Area	FWHM ^1^/eV	Ratio%	Sensitivity factor	Area	FWHM ^1^/eV	Ratio%
@air	5.58	20,379	1.7	63	0.477	1,062	3.7	37
@water		124,318	1.8	46		12,531	3.4	54
@ethanol		301,749	2.1	58		19,131	3.6	42

^1^ FWHM: Full width of Half Maximum

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
