# Peer review of "PL Tunable GaN Nanoparticles Synthesis through Femtosecond Pulsed Laser Ablation in Different Environments"

_nanomaterials, 2020, doi:10.3390/nano10030439_

Round 1

Reviewer 1 Report

This manuscript reports on the fabrication of GaN nanoparticles by femtosecond laser ablation in different environments and the characterization of their photoluminescence.

The manuscript is interesting since the fabrication of nanoparticles by this method presents advantages and could be an alternative to conventional methods. However, the manuscript should be revised, first regarding English grammar. Regarding the content:

  1. Are air, water and ethanol absorbing light at a wavelength of 343 nm? I would say absorption is negligible and thus, that would not be a reason for a different ablation threshold. One aspect to be considered is the focusing: what focal length has the lens used and what is the distance. Is it focusing in front or behind the surface? Then, there could be an effect of the media. Also, considering this, and taking into account the long irradiation time (1 hour) one could expect an effect of the irradiation on the already formed nanoparticles (meaning that the final size of the nanoparticles is indeed a consequence of post irradiation). Have you checked this or can you discuss on this? Besides the viscosity, does the thermal conductivity of the liquid have any effect? See for instance Appl. Surf. Sci. 418, 522 (2017).
  2. Are there significant differences on the amount of nanoparticles obtained? In principle, one of the drawbacks of femtosecond laser ablation is the small amount of material. Any comment on this? 
  3. Regarding stoichiometry it seems that the best nanoparticles are those obtained in water. Are the nanoparticles in the other two cases oxidized? Was this checked (also in comparison to the initial material).
  4. According to the PL spectra shown in figure 5 it seems that there are different contributions in all the spectra, not only in the one corresponding to irradiation in water. The assignment of the peaks should be given and it also seems that the relative intensity of the peaks changes. At least a comment on this should be given.

Reviewer 2 Report

Some minor comments about the article:

1.- Line 26: It has to be included de word EXCITON Bohr radius. Also in other lines below where it is mentioned again.

2.- Line 51: Please correct “rater” and write ratHer.

3.- Line 103: Please correct “structral” and write structUral.

4.- Line 80: In the PLAL process, the fluence (per pulse, in J/cm2), at de focal spot in the target, is much more important than the power. This data should be reported in order to clarify the focusing conditions of the laser beam.

5.- Line 216: The equation about band energy, Eg, is wrong.

6.- Line 217: Only wavelength should be considered (why the product λc?) as the variable linked to Eg variation.

7.- In general, the writing of the article could be improved in order to relate the structural discussion of the nanoparticles to their PL optical properties. In the current writing it could be said that it is a merely descriptive article of well-made Figures.

Round 2

Reviewer 1 Report

Manuscript has been improved and the questions asked during the revision have been answered